# Seed Priming with Single-Walled Carbon Nanotubes Grafted with Pluronic P85 Preserves the Functional and Structural Characteristics of Pea Plants

**DOI:** 10.3390/nano13081332

**Published:** 2023-04-11

**Authors:** Sashka Krumova, Asya Petrova, Nia Petrova, Svetozar Stoichev, Daniel Ilkov, Tsonko Tsonev, Petar Petrov, Dimitrina Koleva, Violeta Velikova

**Affiliations:** 1Institute of Biophysics and Biomedical Engineering, Bulgarian Academy of Sciences, “Acad. G. Bonchev” Str., Bl. 21, 1113 Sofia, Bulgaria; sashka.b.krumova@gmail.com (S.K.); zlatkova.nia@gmail.com (N.P.); svetlio85@abv.bg (S.S.); tsonev@gmail.com (T.T.); 2Institute of Plant Physiology and Genetics, Bulgarian Academy of Sciences, “Acad. G. Bonchev” Str., Bl. 21, 1113 Sofia, Bulgaria; petrova_assya@abv.bg (A.P.); daniel_ilkov@yahoo.com (D.I.); 3Institute of Plant Biology, Biological Research Centre, Temesváry krt. 62, 6726 Szeged, Hungary; 4Institute of Polymers, Bulgarian Academy of Sciences, “Acad. G. Bonchev” Str., Bl. 103, 1113 Sofia, Bulgaria; ppetrov@polymer.bas.bg; 5Faculty of Biology, Sofia University, “St. Kliment Ohridsky”, 1000 Sofia, Bulgaria; koleva@biofac.uni-sofia.bg

**Keywords:** agronanotechnology, carbon nanotubes, chlorophyll fluorescence, leaf anatomy, nanoparticles, photoprotection, photosynthesis, plant biomass, seed germination, seed priming

## Abstract

The engineering of carbon nanotubes in the last decades resulted in a variety of applications in electronics, electrochemistry, and biomedicine. A number of reports also evidenced their valuable application in agriculture as plant growth regulators and nanocarriers. In this work, we explored the effect of seed priming with single-walled carbon nanotubes grafted with Pluronic P85 polymer (denoted P85-SWCNT) on *Pisum sativum* (var. RAN-1) seed germination, early stages of plant development, leaf anatomy, and photosynthetic efficiency. We evaluated the observed effects in relation to hydro- (control) and P85-primed seeds. Our data clearly revealed that seed priming with P85-SWCNT is safe for the plant since it does not impair the seed germination, plant development, leaf anatomy, biomass, and photosynthetic activity, and even increases the amount of photochemically active photosystem II centers in a concentration-dependent manner. Only 300 mg/L concentration exerts an adverse effect on those parameters. The P85 polymer, however, was found to exhibit a number of negative effects on plant growth (i.e., root length, leaf anatomy, biomass accumulation and photoprotection capability), most probably related to the unfavorable interaction of P85 unimers with plant membranes. Our findings substantiate the future exploration and exploitation of P85-SWCNT as nanocarriers of specific substances promoting not only plant growth at optimal conditions but also better plant performance under a variety of environmental stresses.

## 1. Introduction

The process of seed germination is essential for agricultural practices—it needs to be uniform and rapid to achieve the best field performance characteristics and crop yields under different and constantly fluctuating environmental conditions (reviewed in [1]). A number of seed coating and priming techniques are being investigated and exploited to achieve beneficial effects, including recent advancements in agronanotechnology. The exploration of nano-sized agrochemicals for seed treatment is extensively reviewed in [2,3,4,5]. Both positive (plant growth enhancement, environmental safety, improved plant stress resistance) and negative (toxicity, environmental pollution) effects of carbon-based nanopesticides are thoroughly discussed in [5]. The molecular mechanism of nanoparticles action on seeds is not yet clarified; however, it appears that it is dependent on the utilized type of nanoagent, plant species, and mode of interaction between them.

Despite some clues on the effects of nanoparticles, in particular multi-(MWCNT) and single-(SWCNT) walled carbon nanotubes, on plant growth (reviewed in [4,6]), this emerging topic remains largely unexplored. Both MWSCNT and SWCNT are shown to exert positive or negative effects on plants’ physiological states depending on their characteristics (size, surface functionalization, charge, concentration), type of application, plant species, and growth conditions. Bare SWCNT, in particular, have been found to affect *Zea mays*’ root morphology via increased expression of genes involved in seminal root development and decreased expression of genes controlling root hair formation [5]. Root length was also shown to increase upon seed priming with 40 mg/L SWCNT for *Raphanus sativus* and *Brassica rapa* [7], and *Salvia macrosiphon*, *Capsicum annuum*, *Festuca arundinaces* [8]. A wide SWCNT concentration range was tested on *Hyoscyamus niger* seeds by [9], which clearly demonstrated the positive effect on root length and germination percentage up to a concentration of 200 mg/L and an inhibiting effect of higher SWCNT doses.

The study of Cañas et al. [10] demonstrated the strongly individual response of different crops to SWCNT treatments—*Allium cepa* and *Cucumis sativus* seed exposure to non-functionalized SWCNT resulted in enhancement of root length; however, the *Lycopersicon esculentum* root elongation was inhibited under the same treatment. Functionalization of SWCNT with poly-3-aminobenzenesulfonic acid that was utilized to increase the nanoparticles dispersibility resulted in a negative effect on the root length for *Lactuca sativa*, while *Brassica oleracea* and *Daucus carota* were found to be insensitive to the applied treatments [10].

In 2009, Khodakovskaya et al. [11] for the first time showed that MWCNT are able to penetrate the seed coat of tomato seeds and accumulate more water as a result of incubation in Murashige and Skoog medium. The authors suggested that MWCNT enhance the water uptake either by generation of more pores in the seed coat or by regulation of the gating of seed coat water channels (aquaporins). Besides the higher water uptake, the authors also recorded faster germination and higher biomass production over 27 days of seedlings growth. Later on, additional evidence for MWCNT-induced enhancement of plant development are presented by Lahiani et al. [12] and Khodakovskaya et al. [13]. A general trend for increased germination percentage of salvia (*Salvia macrosiphoti* Boiss.), pepper (*Capsicum anntium* L.), and tall fescue (*Festuca arundinacea* Sch.) upon increase in SWCNT concentration was observed by [8]. The authors further revealed that SWCNT are also able to penetrate seed coats and roots. This process depended on the treatment conditions and the cell wall composition in different plant species. SWCNT were also proposed to be able to physically interact with (and penetrate) the cell membranes, including the outer and inner chloroplast membranes, by different mechanisms, such as fluid-phase endocytosis, lipid exchange envelope penetration, and passive diffusion [14,15,16]. They were also suggested to interact with thylakoid membrane lipids [15], pigments [17], and proteins involved in the photosynthetic electron transport [16,18,19,20,21,22,23,24]. Recent studies also demonstrate the possibility for electron leakage and transfer between thylakoids and SWCNT without tight physical interaction with the photosystems [25]. However, the SWCNT–seed interaction is still not thoroughly investigated.

Generally, carbon nanotubes are prone to aggregation, and different strategies to overcome this problem are exploited—polymer-grafting being one of them. To the best of our knowledge, so far only chitosan-wrapped (both covalently and non-covalently bound chitosan) SWCNT with or without additional PEGylation were utilized for successful chloroplast-targeted gene delivery by entering through stomata pores by leaf infiltration [26]. In our previous work, we explored the effect of foliar application of SWCNT grafted by Pluronic P85 (P85) triblock co-polymer (hereon denoted P85-SWCNT) on intact *Pisum sativum* plants [27,28]. We utilized this polymer as a grafting agent for SWCNT since it ensures stable aqueous dispersion of these nanoparticles. We found that P85-SWCNT exerted strong effects on the leaf surface modification and photosynthetic functioning of both photosystem II (PSII) and photosystem I (PSI), which clearly indicated penetration and transportation of this nanomaterial within plant tissues and organelles [27,28]. Interestingly, P85 behaved differently from P85-SWCNT—it had no effect on the examined morphological characteristics; however, when applied in certain concentrations, it led to the increase in some photosynthesis-related parameters, such as maximum efficiency of PSII photochemistry, actual photochemical efficiency of PSII, and fraction of open PSII centers [27]. This prompted us to further explore the application of both P85-SWCNT and P85 in the field of nanoagronomy, this time in regard to seed priming. As far as we are aware, no studies have been performed on pea seed interactions with either SWCNT or MWCNT, although this crop is among the major food resources in Europe [29], and therefore, improvements in its growth technology will have a strong economic impact.

The aim of the present study was to investigate how the initial stages of plant development are affected by seed pre-sawing treatment with P85-SWCNT. We hypothesized that the effect of this priming agent on important agronomical traits, such as seeds germination, plant biomass accumulation, and structural and functional leaf characteristics, is concentration dependent and therefore tested a wide concentration range.

The presented results shed light on the seed/P85-SWCNT interactions and their implications in nanoagriculture. In particular, we demonstrate that P85-SWCNT in concentrations up to 100 mg/L are safe priming agents, i.e., they largely preserve pea leaf physiological characteristics.

## 2. Materials and Methods

### 2.1. Chemicals and Reagents

Poly(ethylene oxide)_26_-block-poly(propylene oxide)_40_-block-poly(ethylene oxide)_26_ triblock copolymer “Pluronic” P85 (used as obtained) was purchased from BASF (Ludwigshafen/Rhein, Germany). SWCNT (>77% carbon) were purchased from Sigma-Aldridge (Darmstadt, Germany) and used without purification. Glutaraldehyde (electron microscopy grade, 25% solution in water) and sodium phosphate buffer were obtained from Thermo Fisher Scientific, Waltham, MA, USA).

### 2.2. Pluronic P85 and P85-SWCNT Dispersions

Polymer suspensions with concentrations of 0.04, 0.2, 1, 5, 10, and 30 g/L were prepared in distilled water, and aliquots were further used for dispersion of 0.4, 2, 10, 50, 100, and 300 mg/L SWCNT (i.e., 100:1 P85:SWCNT *w*/*w* ratio), respectively, upon sonication for 15 min [30]. The polymer solution and P85-SWCNT dispersions were additionally sonicated for 30 min before the start of the seed priming procedure.

### 2.3. Seed Priming

For each individual biological experiment, 50 *Pisum sativum* (var. RAN-1) seeds were soaked in P85 or P85-SWCNT dispersions of the above defined concentrations under continuous shaking at 25 °C for a period of 6 h. Control (hydro-primed) seeds were soaked in distilled water under the same experimental conditions. The seed weights were determined before the start of the imbibition and every 2 h after initial seed soakings. Seed imbibitions (extent of priming mixture components absorption) were determined in relation to the seed dry weights. After the priming procedure, the seeds were re-dried to the original moisture content (which took 10–12 days) and later stored in the dark for an additional 2 days until further use. Before subsequent germination, the primed seeds were soaked in distilled water for 2 h. Figure 1 presents the experimental design steps. Each priming treatment was repeated three times.

### 2.4. Seed Germination and Plant Development

For characterization of the germination process, the primed seeds were placed on wet filter paper in the dark. Root length (threshold of 2 mm) was checked every 24 h for a period of 4 days. The following seed germination parameters as defined in Ranal et al. [31] were determined: germinability (G), i.e., the germination percentage, mean germination time (MT), mean germination rate (MR), and synchrony of the germination process (Z).

On the fourth day of germination, the seeds were placed in containers for hydroponic growth filled with tap water. Plants were grown for an additional 10 days in climatic chamber providing 12 h photoperiod, 400 μmol/m^2^s photon flux density (SMD LED 6500K lamps), 22 °C ambient temperature, and 65% air humidity. At the end of this period, the plants were either used for fresh and dry biomass determination or subjected to anatomical and physiological measurements. The mean % developed plants was estimated by the percentage of developed plants relative to the total number of sown seeds for each variant. The survival index (SVI) was determined by multiplication of the dry weight/plant by the percentage of developed plants at the 14th day of growth for each treatment. Specific leaf area (SLA) and leaf dry mass per unit area (LMA) were determined on fully developed leaves from the second and third leaf pairs. Ten leaf pairs for each individual set of experiments were analyzed. Individual leaf areas were measured using the Image-Pro Plus 6.0 software (Media Cybernetics, Silver Spring, MD, USA). Leaves were dried at 105 °C for 30 min and then at 60 °C to constant weight. LMA was calculated from the ratio of dry leaf biomass and leaf area (g/m^2^).

### 2.5. Leaf Anatomy

Anatomical studies were performed as described in [32]. Briefly, leaf sections (1–2 mm^2^) were collected from the middle part of the second and third well developed leaves and fixed in 3% (*m*/*v*) glutaraldehyde in 0.1 M sodium phosphate buffer, pH 7.4. At least 30 transversal sections per treatment were mounted on slides in glycerol and studied with a light microscope. Images were taken by a digital camera Nikon Eclipse 50i (Nikon Solutions Co., Ltd. Tokyo, Japan). Leaf anatomy was characterized by measuring the thickness of the leaf, palisade and spongy parenchyma, and adaxial and abaxial epidermis.

### 2.6. Leaf Pigments and Chlorophyll Fluorescence

Adaxial surface leaf pigments were measured by the Dualex instrument manufactured by ForceA (Orsay, France). The ratio of chlorophyll (Chl) and flavonoids levels were used to determine the nitrogen balance index (NBI).

Fluorescence imaging was performed by means of IMAGING-PAM MAXI version produced by H. Walz GmbH (Effeltrich, Germany) supplemented with blue excitation light unit (IMAG-MAX/L LED) and IMAG-K7 CCD camera. Prior to the measurements, the plants were dark adapted for 30 min. The experimental setup and the detailed parameter definitions are as in [27]. The efficiency of the photosynthetic apparatus was assessed on the basis of the following parameters: F_v_/F_m_—maximum quantum yield of photosystem II (PSII) (determined in dark-adapted state), Φ_PSII_—quantum efficiency of PSII photochemistry (determined in light-adapted state as in [33]), NPQ—non-photochemical quenching of Chl *a* fluorescence [34]; qL—measure of the fraction of open PSII reaction centers [35]. After the dark adaptation period, leaves were exposed to continuous actinic illumination for 15 min and saturation pulses were applied every 1 min to follow NPQ kinetics. All Chl fluorescence measurements were performed on intact plants.

### 2.7. Statistical Evaluation

Experimental data were subjected to Student’s *t*-test, and three different levels of statistical significance were defined, i.e., *p* < 0.1 (*), *p* < 0.05 (**), *p* < 0.01 (***), as shown in the figure and table captions.

## 3. Results

### 3.1. Seed Germination

As a first step in our study, we checked different periods of hydro-seed priming in the range of 4–24 h. Optimal results for seed imbibition and germination were obtained for 6 h treatment, and all further experiments presented in this work were based on this priming protocol.

Next, we examined the imbibition process in seeds primed with P85-SWCNT and compared the results with those obtained for hydro- (control) and P85-primed ones. For all tested treatments, the imbibition level was similar to the control, and for a period of 6 h it reached ca. 120% from the initial value determined for dry seeds (Table 1).

The germination process was further characterized on the basis of the parameters defined in [31]. For all tested P85-SWCNT concentrations, the mean germinability (G) value was similar to the hydro-primed control. However, this parameter was lower in P85-treated seeds than the hydro-primed control and the respective P85-SWCNT variants, though the differences were not statistically significant (Table 2). The mean time of germination (MT) and mean rate of germination (MR) remained close to the control values for all treatments. The synchrony of germination (Z) was significantly higher than the control only for the 100 mg/L P85-SWCNT treatment (Table 2).

The root development of primed seeds was followed for a period of 4 days. The P85-SWCNT treatment did not lead to changes in root growth relative to the hydro-primed control, while significantly shorter root length was observed in P85 primed seeds (Figure 1).

### 3.2. Plant Development

Plants developed from primed seeds were characterized on the 14th day of their growth at the stage of three well-developed leaf pairs. P85-SWCNT priming did not induce any negative changes in the number of developed plants, LMA, TDB, and SVI as compared to the hydro-primed control, with the sole exception of 100 mg/L concentration that also resulted in lower SVI (Figure 2). For all tested P85 concentrations, the mean % developed plants was lower than the hydro-primed control, but the differences were not statistically significant for the 1 and 5 g/L P85 treatments. The LMA reduced significantly for the 1, 5, and 30 g/L P85 treatments, while all other tested concentrations did not considerably affect this parameter. P85 application (with exception of 10 g/L) led also to largely reduced TBD and SVI values (Figure 2).

### 3.3. Leaf Anatomical Traits

The leaves of control *Pisum sativum* plants were typical bifacial, amphistomatic with an average thickness of the lamina of 268 ± 18 μm (Table 3). The photosynthesizing parenchyma (determined by the distance between adaxial and abaxial epidermis) represented about 80% of the thickness of the lamina. It consisted of a single-row palisade parenchyma and 7–8 rows of spongy parenchyma with a clearly differentiated collecting layer (Appendix A). Epidermal tissues were relatively uniformly structured by tangentially flattened basal epidermal cells and had almost evenly distributed stomata on both surfaces.

Histological examination of the P85-SWCNT and P85 variants showed no anatomical changes for samples primed with low concentrations of P85-SWCNT (0.4, 2 and 10 mg/L) and P85 (0.04, 0.2 and 1 g/L); however, differences in leaf anatomical traits were observed in variants treated with higher concentrations of P85-SWCNT (50, 100 and 300 mg/L) and P85 (5, 10 and 30 g/L). Palisade cells in P85-SWCNT were not typical in shape and had strongly reduced contact, while those in P85 variants were with typically elongated cylindrical shape and with well-defined structural contact ensuring the normal symplast transport (Appendix A). The higher P85-SWCNT concentrations also resulted in significantly decreased leaf thickness (Table 3). The concentration range of 50–300 mg/L P85-SWCNT resulted in lamina thickness decrease of 22–26%, while application of 5–30 g/L P85 resulted in 28–41% reduction. The average thickness of the leaf lamina was reduced by 24% in the P85-SWCNT and 41% in P85-treated variants compared to the hydro-primed control. This was due to the substantial decrease of the spongy parenchyma by 28–30% for P85-SWCNT and 54–64% for P85 variants. No significant changes in palisade parenchyma were detected in either of those variants. As a consequence, the mesophyll thickness decreased by 25–31% and 33–40% for P85-SWCNT and P85 samples, respectively, compared to the control (Table 3). 

### 3.4. Leaf Physiological Characteristics

The total Chl content, assessed on the adaxial leaf surface, remained close to the control one for P85-SWCNT treatments. The Chl level exhibited non-linear P85 concentration dependence—it was significantly higher for 0.04 and 0.2 g/L and lower for 30 g/L P85 variants (Figure 3a). The flavonoid content was lower for 100 mg/L P85-SWCNT and 1 and 30 g/L P85 but higher for 0.4–1 mg/L P85-SWCNT and 10 g/L P85 treatments (Figure 3b). The NBI was significantly higher than the control only for 100 mg/L P85-SWCNT and 1 g/L P85 but was slightly inhibited for 10 g/L P85-primed plants (Figure 3c).

Among all P85-SWCNT treated variants, only 100 mg/L exhibited slightly but significantly higher (at *p* < 0.01) maximum quantum yield of PSII than the control, while this parameter was higher for all P85 treatments, with the exception of 10 g/L (Figure 4a). The Φ_PSII_ parameter increased proportionally to the applied P85-SWCNT amount, but for P85 treatments it was 14–19% higher than the control already at the lowest applied concentration (Figure 4b). A concentration dependence in qL was observed for both P85-SWCNT and P85 treatments (Figure 4c). The steady-state NPQ values reached after 15 min of illumination gradually decreased with the increase in P85-SWCNT concentration, while for all P85 treatments they were largely reduced (by ca. 25%), as compared to hydro-primed seeds (Figure 4d). 

The kinetics of NPQ development followed a similar trend for hydro- and P85-SWCNT-primed samples in the concentration range of 0.4–100 mg/L, i.e., a first phase associated with the initial increase in NPQ values within the first 5 min after illumination and a second one related to attainment of (quasi) steady-state NPQ value after 15 min of illumination. Only for the 300 mg/L P85-SWCNT, the first phase was essentially missing, and the NPQ values for the whole period of 15 min of illumination were lower than in hydro-primed variants (Figure 5). Interestingly, the P85 treatments followed a different course since no clear maximum was visible, and the NPQ values reached steady state already in the first 5 min of illumination; they also remained dramatically lower than the control ones for the tested light period.

## 4. Discussion

### 4.1. Advancements in Nanopriming as New Tool for Germination Improvement

Among the different types of interactions between nanomaterials and plants, the seed priming technique emerges as a widely used approach for improvement of the germination process of a variety of crops [1,3]; however, its further elaboration continues to be of high research interest. The advancements in nanotechnology prompted the researchers to study novel nanomaterials as priming agents; however, this field is still in its infancy. To the best of our knowledge, there are only two reports in the literature regarding the exploitation of carbon nanotubes for this purpose. Yousefi et al. [36] utilized 10–100 mg/L MWCNT as the priming agent for hopbush seeds, which resulted in enhanced germination and biomass accumulation, including in drought stress conditions. However, Lopez-Vargas et al. [37] did not observe any effect on germination after tomato seed priming with graphene and MWCNT (applied in the range 10–1000 mg/L). On the contrary, the authors reported negative impacts on root length and hypocotyl biomass. While 1000 mg/L concentration had a multitude of negative effects, the lower concentrations of the two nanomaterials exhibited distinct and sometimes contrasting effects on a large number of growth-related parameters and enzymatic activities, with strong concentration dependence [37]. The hormesis effect of carbonaceous nanomaterials (effectiveness vs. nanotoxicity) is documented in the literature for different nanomaterials, plant species, and treatment protocols and is recently reviewed in [38].

To gain further fundamental knowledge on the effect of seed priming with carbon nanotubes, in this work, for the first time, we explored the impact of P85-SWCNT on the early stages of pea plant development. It should be noted that in the above-mentioned reports, bare MWCNT were utilized, although this poses a problem with preserving those nanomaterials stably in a non-aggregated state, especially within the plant tissues and organelles. To overcome this obstacle, in our work, we utilized P85-SWCNT (i.e., polymer grafted) that are known to form stable dispersions for a prolonged period. However, since P85 molecules (unimers) are still present in the utilized dispersions, it was also needed to probe their individual action on seeds priming. Although pluronics in general are widely used for biomedical purposes [39], their interactions with seeds and plants are scarcely investigated. To shine some light on this problem, we also characterized the effect of Pluronic P85 as the priming agent. This study is also justified by a number of reports identifying different polymers as beneficial seed coating agents—for example, the study of Samal et al. [40] showed that the coating of cowpea seeds with methyl cellulose, ethyl cellulose, polyvinyl pyrrolidone, hydroxy propyl cellulose, and methyl vinyl acetate polymers results in significantly higher seed germination and better field performance and seed yield per hectare after 6 months of storage in dry conditions. Seeds coating with water absorbent materials is regarded as a novel technique for seeds preservation from drought stress in arid areas [41]. A recent development of polymeric nanofibrous coatings composed of cellulose diacetate is shown to be promising for slow active ingredients delivery to seeds [42].

Our data clearly demonstrate that P85-SWCNT generally appear safe for the plant, at least below 300 mg/L, while P85 exerts strong inhibiting effects, especially on plant biomass accumulation and photoprotection ability. These contrasting effects are further discussed in detail below.

### 4.2. Optimal Conditions for Pea Seed Priming with P85-SWCNT

For the initial 4 days after seeds sowing, we did not detect significant effects of either P85-SWCNT- or P85-priming on seeds imbibition during the 6 h priming procedure and on the germination process. Exploring the plant growth parameters (% developed plants, LMA, TBD, and SVI) of 14 day-old plants developed from P85-SWCNT primed seeds revealed that they are very similar to the hydro-primed ones. The leaf anatomical characteristics (leaf and spongy parenchyma thickness) were significantly affected only for 50–300 mg/L P85-SWCNT treatments; however, the adaxial and abaxial epidermis thickness was preserved as in the control samples. In functional terms, P85-SWCNT primed variants also did not exhibit large deviations as compared to the hydro-primed ones. The total adaxial surface Chl content and Chl fluorescence parameters determined on dark-adapted intact leaves did not vary from the control plants. In light-adapted conditions, however, those variants exhibited higher PSII yield at concentrations above 2 mg/L P85-SWCNT and increased qL values for all tested concentrations, strongly suggesting the presence of more photochemically functional (open) PSII reactions centers with the increase in P85-SWCNT concentration. The NPQ traces recorded for P85-SWCNT-primed variants resembled closely the one of hydro-primed variants, with the exception of 300 mg/L P85-SWCNT.

The fact that P85-SWCNT do not affect seed characteristics but do affect photosynthetic parameters of developed plants strongly suggests that they penetrate the seed without affecting its metabolism and translocate to the photosynthetic organelles where they exert their action. Indeed, numerous authors state that CNT can travel through phloem and xylem and accumulate in different plant organs/organelles, including the chloroplast [4]. The mechanism by which CNT interact with seeds and plant tissues is still unclear; however, two possibilities are proposed—perforation of membranes (i.e., pores formation) and transport via aquaporins [11].

### 4.3. Harmful Effects of Priming with P85 Polymer

P85 treatments exhibited a number of harmful effects at different growth stages—starting from the seed germination (i.e., the root formation stage), followed by thinner lamina and decreased LMA (at concentration ≥ 1 g/L P85), associated with lower biomass accumulation and plants survival and strong inhibition of NPQ (both the initial phases of its development and in its (quasi)steady-state value after 15 min of illumination), which strongly influences the capability of plants to cope with light stress. The lower LMA and leaf lamina thickness is believed to be related to lower physical strength and shorter leaf lifespans [43,44,45], but it also relates to photosynthesis efficiency via allocation of nitrogen within photosynthetic tissue, CO_2_ diffusion regulation, and light penetration within photosynthetic tissues [45]. Based on our results, we assume that the lower LMA in P85-treated variants results in higher photon flux reaching the photosynthetic apparatus, which leads to higher PSII quantum yield. This statement is supported by the fact that the arrangement of palisade cells in P85 plants should in principle facilitate light channeling deeper into the leaf [46,47,48,49]. However, the decrease in mesophyll thickness (due to lower spongy parenchyma) in P85-primed samples must affect the light scattering and focusing within the leaf as well as the gas conductivity processes and is probably associated with lower tolerance for high light intensity [48,50]. In a number of species, lower mesophyll thickness is also correlated with larger capacity for chloroplast movement—another tool to cope with short-term light intensity fluctuations [51]. Furthermore, the thinner epidermis in P85 variants is likely to reduce the leaf protection from the harmful UV-B rays [48]. All those features have long-term consequences for the plant growth of P85 variants as well as for their photosynthetic and photoprotection function since thinner mesophyll was also shown to be related to incomplete development of the NPQ process [52,53]. The data clearly show that in P85-primed variants, a large portion of the PSII excitation energy is not safely dissipated as heat (a major mechanism for photoprotection) but emitted as fluorescence (hence the high PSII quantum yield). Furthermore, those variants appear to be trapped in a partially photoinhibited state that does not allow for full development of the NPQ process. Thus, it could be expected that P85 priming will enhance the plant susceptibility to high light exposure and consequently photooxidative damages to the photosynthetic apparatus, which however requires further experimental verification. Recent studies showed that the time course of NPQ is related to biomass accumulation [54,55,56]. Indeed, in our study we found a similar correlation—a significant reduction in biomass accumulation is accompanied with substantially lower NPQ values in P85 variants.

The variety of negative effects of P85 priming can be explained with the low size of P85 unimers that can easily transverse the seed coat and cell wall, and enter the plant cells. It is well known that due to their amphiphilic nature pluronic unimers and micelles do interact with biological membranes and modify their properties (reviewed in [56]).

## 5. Conclusions

Achieving sustainable agriculture is a major goal for the present and future generations since there is an urgent need for reduction of environmental pollution, sustainable utilization of the natural resources, development of crops that would resist climate change challenges, and optimization of food production and price [57].

Here, we explored a number of growth- and physiology-related parameters as markers for the effect of pea seed priming with Pluronic P85 tri-block copolymer grafted single-walled carbon nanotubes (P85-SWCNT). Detailed exploration of the seed germination process, plant biometry characteristics, leaf anatomy, physiology, and photosynthetic activity revealed that P85-SWCNT seed priming in the concentration range of 0.4–100 mg/L is safe for pea plant development and even stimulates photosystem II photochemistry. Therefore, P85-SWCNT nanoparticles appear as suitable object for further development as nanocarriers of specific substances acting as plant growth regulators. Importantly, anatomical and physiological changes occurring in P85-SWCNT- and P85-treated plants revealed that those two agents have distinct modes of action. On one hand, it appears that P85-SWCNT application supports plants to largely overcome the negative impact of the P85 polymer. On the other hand, however, the fact that a large amount of P85 unimers is in a bound form in P85-SWCNT particles might be the reason for the much smaller negative effect exerted by the free P85 unimers in P85-SWCNT dispersion. Nevertheless, the evaluation of P85-SWCNT toxicity, in particular, requires a further dedicated study.

## Data Availability

The data are contained within the article and the Appendix A.

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
