# Peer review of "Seed Priming with Single-Walled Carbon Nanotubes Grafted with Pluronic P85 Preserves the Functional and Structural Characteristics of Pea Plants"

_nanomaterials, 2023, doi:10.3390/nano13081332_

Round 1

Reviewer 1 Report

The manuscript entitled "Seeds priming with single-walled carbon nanotubes grafted with Pluronic P85 preserves the functional and structural characteristics of pea plants" reports a research on single-walled carbon nanotubes grafted with the PluronicP85 polymer, that was  used and studied in the growth of seeds of pea (Pisum -sativum). The manuscript is well written, the subject matter is interesting.

I would like to send only one observation to the authors: A greater explanation should be given on how and by which mechanisms carbon nanotubes manage to improve the parameters that have been studied. What is actually the role played by  carbon nanotubes. More explanations or hypotheses would be needed

Author Response

Comments

The manuscript entitled "Seeds priming with single-walled carbon nanotubes grafted with Pluronic P85 preserves the functional and structural characteristics of pea plants" reports a research on single-walled carbon nanotubes grafted with the PluronicP85 polymer, that was  used and studied in the growth of seeds of pea (Pisum -sativum). The manuscript is well written, the subject matter is interesting.
I would like to send only one observation to the authors: A greater explanation should be given on how and by which mechanisms carbon nanotubes manage to improve the parameters that have been studied. What is actually the role played by  carbon nanotubes. More explanations or hypotheses would be needed.

ANSWER:

We thank Reviewer 1 for the positive opinion about our manuscript. We took into account his/her suggestion. In the revised version of the manuscript we provide more information about the possible mechanisms of action of SWCNT in plants (lines 91-98, 512-519).  Some аssumptions about the possible mode of action of the SWCNT on the photosynthetic apparatus  have been proposed in our previous studies where leaves were sprayed with SWCNT and therefore the probability of their interaction with photosynthetic membranes is higher (Velikova et al. International Journal of Molecular Sciences 22(9):4878, 2021. https://doi.org/10.3390/ijms22094878; Petrova et al., Molecules 26(19):5958, 2021. https://doi.org/10.3390/molecules26195958). Based on the results reported in the present study, however, it is difficult to hypothesize any mechanism of action especially on seed germination and plant growth. It should be taken into account that upon seed priming the SWCNT must overcome a number of barriers (cell walls, cellular membranes, phloem, extracellular spaces) before reaching the photosynthetic apparatus (where we observe the strongest effects). Those mechanisms are far beyond our goals. Undoubtedly, however, such studies would be of great interest to scientists working in the field.

Reviewer 2 Report

I have received the Manuscript (Manuscript Number: nanomaterials-2311334) entitled: ‘Seeds priming with single-walled carbon nanotubes grafted with Pluronic P85 preserves the functional and structural characteristics of pea plants’ submitted to Nanomaterials for a review.

This is very well prepared Manuscript deprived of weaknesses or errors decreasing its quality. The scientific hypothesis is well-defined and the methodology regarding experimental part used in this Manuscript are adequate and clear. By analysis of discussion section, it can be concluded that the authors took care with data analysis. My only one advice is to give full names of all abbreviations in conclusions, due to the fact that majority of scientists start reading the Manuscript form this very section.

Author Response

Reviewer 2

Comments

 I have received the Manuscript (Manuscript Number: nanomaterials-2311334) entitled: ‘Seeds priming with single-walled carbon nanotubes grafted with Pluronic P85 preserves the functional and structural characteristics of pea plants’ submitted to Nanomaterials for a review.

This is very well prepared Manuscript deprived of weaknesses or errors decreasing its quality. The scientific hypothesis is well-defined and the methodology regarding experimental part used in this Manuscript are adequate and clear. By analysis of discussion section, it can be concluded that the authors took care with data analysis.

My only one advice is to give full names of all abbreviations in conclusions, due to the fact that majority of scientists start reading the Manuscript form this very section.

ANSWER:

 We thank Reviewer 2 for the positive opinion about our manuscript. We took into account his/her suggestion and included the full names of the abbreviations in the Conclusions section (lines 579-598).

Reviewer 3 Report

The manuscript titled “Seeds priming with single-walled carbon nanotubes grafted with Pluronic P85 preserves the functional and structural characteristics of pea plants” was aimed to investigate how the initial stages of plant development are affected by seeds pre-sawing treatment with P85-SWCNT. We hypothesized that the effect of this priming agent on important agronomical traits such as seeds germination, plant biomass accumulation, and structural and functional leaf characteristics is concentration dependent and therefore tested wide concentration range. The topic is relevant and the work can be interesting for readers, however presented manuscript needs to be revised. Some comments are listed below:

1. Abstract should be reached by the main important data obtained.

2. Nanomaterials can as stimulate seeds germination as suppress it causing stress via various mechanisms (https://doi.org/10.3390/coatings11070862, https://doi.org/10.1038/s41598-020-70303-8). The reverse side of nanomaterials application in crop processing should be mentioned in Introduction.

3. A scheme of the experiment is needed in Material and Methods section

4. Chemicals and reagents used better be mentioned in separate subsection in Material and Methods section with given details (chemical grade, manufacturer, city, country).

5. For equipment and soft details should be unified (manufacturer, city, country).

6. The authors should explain why the seeds were shaken in solution for 6 h.

7. There are many old references in Discussion section. The authors should focus on recent works and compare their results with relevant studies of other researchers. For example, https://doi.org/10.1016/j.ecoenv.2017.09.024

8. Conclusion should be modified. More important data obtained should be showed and discussed. Future prospects and industrial application should be described as well.

Author Response

Reviewer 3

Comments

The manuscript titled “Seeds priming with single-walled carbon nanotubes grafted with Pluronic P85 preserves the functional and structural characteristics of pea plants” was aimed to investigate how the initial stages of plant development are affected by seeds pre-sawing treatment with P85-SWCNT. We hypothesized that the effect of this priming agent on important agronomical traits such as seeds germination, plant biomass accumulation, and structural and functional leaf characteristics is concentration dependent and therefore tested wide concentration range. The topic is relevant and the work can be interesting for readers, however presented manuscript needs to be revised. Some comments are listed below:

  1. Abstract should be reached by the main important data obtained.

ANSWER:

We thank Reviewer 3 for the critical comments, which we consider relevant and needed in order to improve our work. According to the Reviewer’s requirement, we have revised the abstract and included more specific details on the obtained results.

 2. Nanomaterials can as stimulate seeds germination as suppress it causing stress via various mechanisms (https://doi.org/10.3390/coatings11070862, https://doi.org/10.1038/s41598-020-70303-8). The reverse side of nanomaterials application in crop processing should be mentioned in Introduction.

ANSWER:

We added refs regarding the positive and negative effects of nanomaterials (organic and inorganic) on seeds. We would like to point out that there is enormous amount of data related to the application of various types of nanoparticles on various plant species, utilizing different treatment protocols. In order to keep the Introduction concise are relevant to our experimental work, we limited the referenced articles to the specific effects of carbon nanotubes observed on seeds, plants and photosynthetic membranes.

  1. A scheme of the experiment is needed in Material and Methods section.

ANSWER:

A scheme of the experimental design is provided in the revised version of the ms (Scheme 1 in Material and Methods section). 

  1. Chemicals and reagents used better be mentioned in separate subsection in Material and Methods section with given details (chemical grade, manufacturer, city, country).

ANSWER:

A separate subsection in  Material and Methods (2.1. Chemicals and reagents) now is added, where details about chemicals and reagents used are provided.

  1. For equipment and soft details should be unified (manufacturer, city, country).

ANSWER:

We corrected that in the revised version of the manuscript.

  1. The authors should explain why the seeds were shaken in solution for 6 h.

ANSWER:

In our preliminary experiments we have checked different periods of hydro-seed priming in the range 4 - 24 h. Optimal results were obtained for 6h treatment, therefore all further experiments were based on this priming protocol. This clarification is now added in the beginning of Results section (line 262-265).

  1. There are many old references in Discussion section. The authors should focus on recent works and compare their results with relevant studies of other researchers. For example, https://doi.org/10.1016/j.ecoenv.2017.09.024.

ANSWER:

The interaction between SWCNT and seeds is still not thoroughly investigated, and the information is rather scarce. We added new relevant studies focused on carbon nanotubes and their possible mode of action in plants (ref. 2, 5, 15, 17, 23-25, published in the last 5 years).

  1. Conclusion should be modified. More important data obtained should be showed and discussed. Future prospects and industrial application should be described as well.

ANSWER:

We modified the Conclusion section according to the Reviewers suggestion.

Round 2

Reviewer 3 Report

The authors considered all comments and decided them well. the revised manuscript can be accepted for publication in Nanomaterials